# DeepDefense: Layer-Wise Gradient-Feature Alignment for Building Robust Neural Networks

## Abstract

Deep neural networks are known to be vulnerable to adversarial perturbations—small, carefully crafted inputs that lead to incorrect predictions. In this paper, we propose *DeepDefense*, a novel defense framework that applies Gradient-Feature Alignment (GFA) regularization across multiple layers to suppress adversarial vulnerability. By aligning input gradients with internal feature representations, DeepDefense promotes a smoother loss landscape in tangential directions, thereby reducing the model's sensitivity to adversarial noise.

We provide theoretical insights into how adversarial perturbation is decomposed into radial and tangential components and demonstrate that alignment suppresses loss variation in tangential directions, where most attacks are effective. Empirically, our method achieves significant improvements in robustness across both gradient-based and optimization-based attacks. For example, on CIFAR-10, CNN models trained with DeepDefense outperform standard adversarial training by up to 15.2% under APGD attacks and 24.7% under FGSM attacks. Against optimization-based attacks like DeepFool and EADEN, DeepDefense requires 20-30 times higher perturbation magnitudes to cause misclassification, indicating stronger decision boundaries and flatter loss landscape. Our approach is architecture-agnostic, simple to implement, and highly effective, offering a promising direction for improving the adversarial robustness of deep learning models.

## 1 Introduction

Deep Neural Networks (DNNs) have achieved remarkable success in a wide range of applications, including object recognition (Zou et al., 2023), natural language processing (Zhou et al., 2024), and associative memory (Lin et al., 2024b; 2023). Despite their high performance, DNNs are vulnerable to adversarial attacks, imperceptible input perturbations carefully crafted to mislead models into incorrect predictions (Nguyen et al., 2015). This vulnerability poses a critical threat to the deployment of deep learning in high-stakes applications, including medical diagnosis (Javed et al., 2025), autonomous driving (Ahmed et al., 2025), agricultural forecasting (Lin et al., 2024a), and cybersecurity (Al Siam et al., 2025).

In recent years, substantial progress has been made in developing adversarial defense strategies. A prominent class is adversarial training, where models are trained on adversarial examples to enhance robustness (Shafahi et al., 2019). Variants such as tradeoff-inspired Adversarial Defense via Surrogate-loss minimization (TRADES) (Zhang et al., 2019) and Fast Adversarial Training (Wong et al., 2020) offer improved trade-offs between performance and computational efficiency. Meanwhile, certified defenses aim to provide provable guarantees against attacks within bounded perturbations (Levine & Feizi, 2020). Techniques like randomized smoothing (Cohen et al., 2019) and interval bound propagation (IBP) (Gowal et al., 2018) have shown promise in offering formal robustness certificates.

Another line of defense involves input transformation methods, which seek to sanitize or denoise potentially adversarial inputs before classification. Examples include feature denoising networks (Xie et al., 2019a) and denoising diffusion models (Nie et al., 2022). In parallel, efforts to mitigate gradient obfuscation—a common pitfall in poorly designed defenses—have led to approaches like ensemble

adversarial training (Tramèr et al., 2018) and backdoor-resilient architectures (Liu et al., 2018a). More recently, researchers have begun designing defenses that target the internal representations and architecture of deep networks. These include Lipschitz-constrained models (Tsuzuku et al., 2018; Gouk et al., 2021), gradient alignment regularization (Ross & Doshi-Velez, 2018), and robust Vision Transformers (ViTs) (Mao et al., 2022). Finally, post-hoc detection mechanisms attempt to identify adversarial inputs after inference, using methods such as Mahalanobis distance-based anomaly detection (Lee et al., 2018) and logit-space inconsistency checks (Dathathri et al., 2018).

## 1.1 MOTIVATION AND CONTRIBUTION

Existing defense methods, such as adversarial training, feature denoising, randomization, and distillation, either rely on expensive retraining or focus on isolated layers, often failing to prevent perturbation propagation through the network. These approaches treat robustness as a surface-level property, overlooking the internal dynamics of adversarial noise. To address this, we propose DeepDefense, which blocks adversarial perturbations at every layer by enforcing Gradient-Feature Alignment (GFA). This layer-wise regularization suppresses distortion propagation, offering a scalable and intrinsically robust defense.

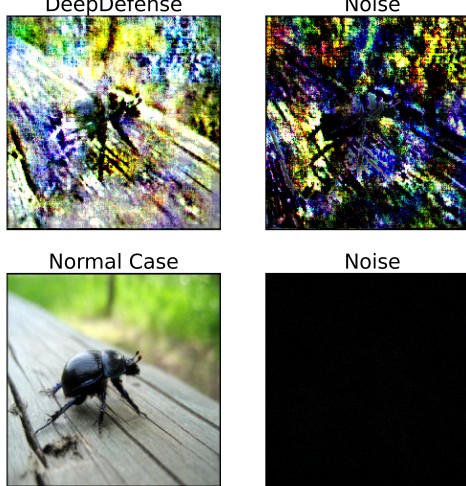

Figure 1: Adversarial examples and the corresponding noise generated by DeepFool. A model trained with DeepDefense (top) requires larger, more visible perturbations to fool than a standard model (bottom), demonstrating greater robustness.

As shown in Figure 1, for most optimization-based attack approaches, such as DeepFool, a large amount of noise (which is perceptible) along the radial direction of the input sample is required to mislead the CNN trained with the DeepDefense strategy. In contrast, the model trained using standard backpropagation can be misled by adding an imperceptible amount of noise. DeepDefense employs layer-wise GFA regularization to ensure that adversarial perturbations do not accumulate as they propagate from early convolutional layers to deeper fully connected layers, thereby increasing the model's resistance to attacks. The main contributions of this study are as follows:

- Introduce Gradient-Feature Alignment (GFA) regularization across different layers as a simple yet effective intrinsic defense that aligns input gradients with feature representations to promote a flatter loss landscape and enhance robustness.
- Provide theoretical and empirical analysis showing that adversarial perturbations can be decomposed into radial and tangential components relative to the input. GFA suppresses vulnerability along tangential directions—where most attacks are effective—while noise in the radial direction is naturally filtered by the model, resulting in improved robustness.
- Demonstrate enhanced robustness of GFA-trained models across a wide range of adversarial attacks, including both gradient-based and optimization-based methods, with visual and quantitative evidence confirming improved decision boundaries.

Through extensive experiments, we demonstrate that DeepDefense significantly enhances model robustness, outperforming existing adversarial defense techniques.

## 1.2 ORGANIZATION

The remainder of this paper is organized as follows. Section 2 reviews related adversarial defense strategies, with a focus on gradient alignment methods. Section 3 presents the mathematical formulation of our proposed Gradient-Feature Alignment (GFA) regularization and its layer-wise extension, DeepDefense. Section 4 describes the experimental setup and evaluates the performance of DeepDefense across various adversarial attacks on CNN and MLP models. Section 5 concludes the paper

and outlines directions for future work. Additional implementation details and extended results are provided in the Appendix.

## 2    RELATED WORK

Numerous defense strategies have been proposed to mitigate the vulnerability of deep learning models to adversarial examples. One of the most widely adopted techniques is *adversarial training*, where models are trained on adversarially perturbed inputs to improve robustness (Bai et al., 2021; Wang et al., 2024). Although effective, adversarial training is computationally expensive and often overfits to specific attack types, limiting generalizability (Tramèr et al., 2017).

To address these challenges, *randomization-based defenses* have been introduced. These methods incorporate stochastic components into models, such as random input transformations or noise injection into intermediate activations (Cohen et al., 2019; Liu et al., 2018b; Dhillon et al., 2018). The goal is to obscure the gradient path and reduce the transferability of adversarial attacks. However, many such defenses have been defeated by *adaptive attacks*, highlighting the need for more principled approaches (Athalye et al., 2018).

Another active line of research involves *ensemble-based defenses*, which aim to improve robustness by leveraging the diversity of multiple models (Liu et al., 2019; Pang et al., 2019). These methods often promote diversity through variations in model architecture, decision boundaries, or output behavior. For example, *Adaptive Diversity Promoting* (ADP) training explicitly encourages disagreement among ensemble members to reduce the risk of shared vulnerabilities (Pang et al., 2019). In addition, *Gradient Alignment Loss* (GAL) has been proposed to further improve ensemble robustness by penalizing excessive gradient alignment, thereby reducing the shared adversarial subspace and lowering the transferability of attacks across ensemble members (Pang et al., 2019). However, ensemble defenses consume significant computational resources and are generally heavy and inefficient.

Consequently, another family of defense mechanisms has emerged that improves the robustness of a single network by leveraging gradient information. For example, *Gradient Norm Regularization* (GNR) penalizes the magnitude of the input gradients, aiming to control the sensitivity of the model to small input perturbations (Ross & Doshi-Velez, 2018). However, upon deeper analysis, it is found that GNR has limited effect on improving the robustness of neural networks. In addition, *Gradient Alignment Regularization* (GAR) methods introduce additional constraints informed by external knowledge or auxiliary models. For instance, *Perceptual Alignment of Gradients* (PAG) aligns a model's input gradients with perceptually meaningful directions derived from robust or pre-trained models (Ganz et al., 2023). *Input Gradient Alignment Matching* (IGAM) is another example, designed for student-teacher frameworks. It transfers robustness and interpretability from a robust teacher model to a student model by aligning their input gradients during training (Chan et al., 2020). Similarly, *Gradient Alignment with Image Edges* (GAIE) promotes alignment between input gradients and edge features extracted using image processing techniques, encouraging the model to focus on structurally significant regions of the input (Rodríguez-Muñoz et al., 2024). Furthermore, *GradAlign* builds upon these ideas in the context of adversarial training by enforcing consistency of input gradients across perturbed samples. This stabilizes single-step adversarial training methods, such as FGSM, and mitigates the problem of catastrophic overfitting (Ross & Doshi-Velez, 2018).

However, although these gradient alignment methods are effective, they require either a robust neural network or a pre-trained teacher model, which limits their practicality in real-world scenarios. Our proposed method, *DeepDefense*, builds on the foundation of GFA regularization by introducing a layer-wise regularization strategy that aligns the input to each layer with its corresponding input gradient, offering a self-contained mechanism that modulates internal representations according to their gradient flow, without requiring perceptual priors or robust teacher models.

## 3    MATHEMATICAL FORMULATION FOR DEEPDEFENSE FRAMEWORK

### 3.1    GRADIENT-FEATURE ALIGNMENT REGULARIZATION

As its name suggests, GFA regularization enforces the alignment between the input vector $\mathbf{x}$ and the input gradient $\nabla_{\mathbf{x}}\mathcal{L}$, where $\mathcal{L}$ is the loss function, using cosine similarity, as shown in Equation 1.

$$\text{GFA} : \cos\theta = \frac{\langle \mathbf{x}, \nabla_{\mathbf{x}}\mathcal{L} \rangle}{\|\mathbf{x}\| \cdot \|\nabla_{\mathbf{x}}\mathcal{L}\|} \tag{1}$$

When the gradient of the loss with respect to the input (or feature) is aligned with the input itself, the loss varies predominantly in the radial direction (along the input vector), while remaining nearly constant in tangential directions (orthogonal to the input).

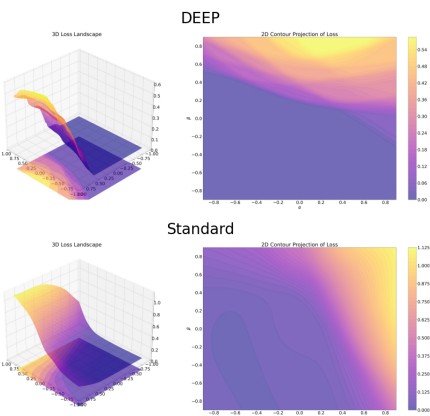

Figure 2: Loss landscapes of models trained with the DEEP strategy (top) and standard backpropagation (bottom), shown in both 2D and 3D. The DEEP model exhibits smoother surfaces with overall flatness, while the standard model is relatively flat only in the radial direction and more sensitive to perturbations.

As shown in Figure 2, the perturbation directions are generated using APGD from models trained with and without GFA, respectively. We combine perturbations from both directions in varying ratios and visualize the loss landscapes for two models: one trained with standard backpropagation and the other with GFA regularization. It is evident that the model trained with GFA regularization exhibits a much flatter loss landscape compared to the standard model. This is because, in the GFA-regularized model, perturbations in the radial direction are largely suppressed by activation functions or the inherent structure of the neural network, while perturbations in the tangential direction do not significantly increase the loss value, as discussed below.

### 3.1.1 ORTHOGONAL VERSUS RADIAL GRADIENT BEHAVIOR

In a model trained with standard backpropagation, the gradient $\nabla_x L(x)$ empirically tends to be approximately orthogonal to the input $x$: $\nabla_x L(x)^\top x \approx 0$. As a result, the loss changes most significantly in directions perpendicular to $x$. For a perturbation $\delta$ satisfying $\delta^\top x = 0$, we have: $D_\delta L(x) = \nabla_x L(x)^\top \delta \approx \|\nabla_x L(x)\| \cdot \|\delta\|$. Consequently, the loss landscape exhibits high curvature in tangential directions, making models particularly sensitive to adversarial perturbations orthogonal to the input $x$.

However, GFA regularization enforces that $\nabla_x L(x) = \lambda x$. This implies that the loss varies only along the radial direction of $x$ and remains invariant in all tangential directions. For any $\delta \perp x$, we have: $D_\delta L(x) = \nabla_x L(x)^\top \delta = \lambda x^\top \delta = 0$. This results in a loss surface that is flat in tangential directions, forcing adversarial attackers to search along the radial direction to mislead the model. However, radial perturbations are inherently suppressed due to activation function damping or cancellation by normalization layers.

### 3.2 DEEPDEFENSE: LAYER-WISE APPLICATION OF GFA

Let $f : \mathbb{R}^d \to \mathbb{R}^K$ be a deep neural network composed of $L$ layers, where the input is $\mathbf{x} \in \mathbb{R}^d$ and the output is a classified vector in $\mathbb{R}^K$. Each layer $l \in \{1, \ldots, L\}$ has an input activation $\mathbf{h}^{(l)}$ and produces a feature representation $\mathbf{z}^{(l)}$. DeepDefense enforces robustness by applying GFA regularization at a variety of layers. For each layer $l$, we compute the gradient of the loss function $\mathcal{L}$ with respect to the layer input $\mathbf{h}^{(l)}$: $\nabla_{\mathbf{h}^{(l)}} \mathcal{L} = \frac{\partial \mathcal{L}}{\partial \mathbf{h}^{(l)}}$.

To suppress perturbation propagation, we enforce alignment between the gradient direction and the layer's feature representation using cosine similarity, as shown in Equation 1. The GFA loss is defined as the negative cosine similarity (to encourage alignment), and the total regularized loss is shown in Equation 2.

$$\mathcal{L}_{\text{total}} = \mathcal{L}_{\text{task}} + \beta \sum_{l=1}^{L} \left( 1 - \text{GFA}^{(l)} \right) \tag{2}$$

where $\beta$ is a hyperparameter vector controlling the strength of the GFA regularization at different layers.

By applying this penalty across different layers, DeepDefense prevents adversarial perturbations from accumulating or amplifying through the network, thereby enhancing robustness of models. The pseudoscope of the deep defense algorithm is shown in Algorithm 1.

---

**Algorithm 1** Deep Defense Algorithm

---

**Require:** Training data $D = \{(x_i, y_i)\}_{i=1}^N$, learning rate $\eta$, model parameters $\theta$, loss coefficients $\alpha, \beta$.
1: Initialize Deep Learning Model Parameters $\theta$
2: **for** each batch $(X, Y)$ in $D$ **do**
3:     $(\hat{Y}, F) \leftarrow f_\theta(X)$
4:     $L_{\text{MSE}} \leftarrow \text{MSE}(\hat{Y}, Y)$
5:     **for** each feature $F_i \in F$ **do**
6:         $\nabla_{F_i} L_{\text{MSE}} \leftarrow \frac{\partial L_{\text{MSE}}}{\partial F_i}$
7:         $\text{GFA}_i \leftarrow \frac{\nabla_{F_i} L_{\text{MSE}} \cdot F_i}{\|\nabla_{F_i} L_{\text{MSE}}\|\|F_i\|}$
8:     **end for**
9:     $L \leftarrow \alpha L_{\text{MSE}} - \beta * \text{GFA}$
10:     $\theta \leftarrow \theta - \eta \nabla_\theta L$
11: **end for**

---

### 3.2.1 INTER-LAYER INFLUENCE OF GFA REGULARIZATION

Assuming an $n$-layer neural network, the forward propagation is shown in Equation 3.

$$z_i = W_i a_{i-1}, \quad a_i = \sigma(z_i), \quad \text{for } i = 1, 2, ..., n \tag{3}$$

where $x = a_0$ is the input, $W_i$ are the weight matrices at the layer $i$, $\sigma(\cdot)$ is the activation function, and $a_n$ is the final output used for loss computation.

Then we can compute the gradient of the input in each layer by backpropagation and chain rule, the gradient with respect to the output layer ($i = n$) are shown in Equation 4.

$$\frac{\partial L}{\partial z_n} = \frac{\partial L}{\partial a_n} \odot \sigma'(z_n), \quad \frac{\partial L}{\partial a_{n-1}} = \frac{\partial L}{\partial z_n} W_n \tag{4}$$

Through recursive, we can compute the gradient of the input in each layer $i$, as shown in Equation 5.

$$\frac{\partial L}{\partial a_i} = \left( \left( \frac{\partial L}{\partial a_n} \odot \sigma'(z_n) \right) W_n \odot \sigma'(z_{n-1}) \right) \\ W_{n-1} \cdots \odot \sigma'(z_{i+1}) W_{i+1} \tag{5}$$

Therefore the input gradient of the n layer neural network are shown in Equation 6.

$$\frac{\partial L}{\partial x} = \prod_{i=1}^n (\sigma'(z_i) W_i) \frac{\partial L}{\partial a_n} \tag{6}$$

From Equation 6, it is straightforward to observe that the input gradient of the layer $i$ impact the input gradient of the layer $i + 1$. Therefore, in the following experiment, although we only force the GFA at specified layers, we will record the GFA values for all layers for comparison and discussion purposes.

## 4 EXPERIMENT AND DISCUSSION

### 4.1 EXPERIMENT CONFIGURATION

To evaluate the effectiveness of our proposed defense mechanism, DeepDefense, we conduct experiments on CIFAR-10 using Convolutional Neural Network (CNN) architecture. All input samples

Table 1: GFA Regularization Values for Models Trained with Different Strategies[1]

| Strategies | Dataset | GFA Regularization Value | | | | |
|---|---|---|---|---|---|---|
| CNN | train | -0.0003 ± 0.0009 | -0.0006 ± 0.0005 | 0.0 ± 0.0006 | 0.0 ± 0.0011 | 0.0014 ± 0.0023 |
| | test | 0.0029 ± 0.0003 | 0.0004 ± 0.0004 | 0.003 ± 0.0003 | 0.0065 ± 0.0005 | 0.0116 ± 0.0014 |
| ADV | train | 0.0008 ± 0.0035 | -0.0079 ± 0.0081 | 0.0004 ± 0.0013 | -0.0012 ± 0.0016 | 0.0048 ± 0.0045 |
| | test | 0.0009 ± 0.0014 | -0.0062 ± 0.0035 | 0.001 ± 0.0004 | -0.0022 ± 0.0006 | -0.0019 ± 0.0019 |
| FIRST | train | 0.9449 ± 0.0041 | 0.4207 ± 0.027 | -0.0008 ± 0.0002 | -0.0104 ± 0.0045 | -0.0473 ± 0.0896 |
| | test | 0.4734 ± 0.0112 | 0.1955 ± 0.0166 | -0.0005 ± 0.0002 | -0.0059 ± 0.0027 | -0.0257 ± 0.0545 |
| DEEP | train | 0.9262 ± 0.0039 | 0.9034 ± 0.013 | 0.9448 ± 0.0039 | -0.0077 ± 0.0025 | -0.0954 ± 0.0243 |
| | test | 0.5409 ± 0.0396 | 0.5173 ± 0.038 | 0.5395 ± 0.042 | -0.005 ± 0.0022 | -0.0554 ± 0.01 |
| GAIE[2] | train | -0.103 ± 0.0207 | -0.0035 ± 0.0287 | -0.0006 ± 0.0011 | 0.0052 ± 0.0031 | -0.078 ± 0.0906 |
| | test | -0.0262 ± 0.012 | 0.0035 ± 0.0161 | 0.0 ± 0.0007 | 0.0037 ± 0.0018 | -0.0469 ± 0.0582 |

[1] Although GFA regularization is explicitly applied only in **FIRST** and **DEEP**, GFA values are calculated for all models to evaluate the impact of GFA on model robustness. On the training dataset, the **FIRST** and **DEEP** models exhibit similar GFA values (around 0.93), and the GAIE regularization value for **GAIE** is also approximately 9.3, enabling a fair comparison among these three models. Moreover, this value nearly reaches the limit of the model when the corresponding regularization is applied. However, on the testing dataset, GFA values differ significantly from those in the training dataset, suggesting that GFA may not fully generalize to unseen data in CIFAR-10 due to the presence of noisy information and the fact that the model did not learn relevant features that could be useful for classifying the testing samples. To mitigate this issue, using data augmentation to generate more training samples may help reduce this discrepancy.

[2] The GAIE loss for this model is $0.9360 \pm 0.0141$ on the training dataset, while on the testing dataset, it is $0.4516 \pm 0.0278$.

are rescaled to the range [-1, 1] prior to training and evaluation. The models are optimized using the mean squared error (MSE) loss function.

All experiments are executed on an Ubuntu system equipped with an NVIDIA GeForce RTX 3090 GPU (25 GB memory), ensuring efficient model training and robust evaluation. Each model is trained using the corresponding training dataset, and robustness is assessed on the testing dataset. Importantly, to ensure a fair and consistent evaluation of adversarial robustness, only the test samples that are correctly classified by the all models are included in the attack phase. Unless otherwise specified, all adversarial attacks are conducted using identical configurations across different models.

## 4.2 CASE STUDY: CONVOLUTIONAL NEURAL NETWORKS ON CIFAR-10

To evaluate our approach, we conduct a case study using the CIFAR-10 dataset. Six CNN models are developed, each trained with a different strategy, and evaluated under a consistent configuration against multiple adversarial attackers. For clarity, we refer to each model using the following shorthand labels:

- **CNN**: Standard training using clean CIFAR-10 data.
- **ADV**: Adversarial training with a mix of clean and PGD-generated adversarial examples.
- **FIRST**: GFA regularization applied to the first convolutional layer.
- **DEEP**: GFA regularization applied to the first three layers.
- **GAIE**: GAIE regularization applied to the first convolutional layer.
- **DENOISE**: Feature-denoising layers inserted into the first and second convolutional layers.

For the models trained with GFA (**FIRST** and **DEEP**) as well as **GAIE**, we ensure that the corresponding regularization losses applied to the first layer are maintained at a similar scale (around 0.93) on the training dataset, as shown in Table 1. Moreover, these values nearly reach the limit of the models on the training dataset when the corresponding regularization is applied. However, GFA did not fully generalize to the testing dataset, as shown in Table 1, likely due to the presence of noisy background information and the fact that the model did not learn relevant features that are useful for classifying the testing samples. It is beneficial to use a data augmentation approach to reduce this discrepancy. This design choice enables a fair comparison of the effectiveness of various strategies in improving robustness. Furthermore, although **ADV** is not explicitly trained with GFA, its GFA on

the training dataset is slightly higher than that of the model trained with standard backpropagation. However, this trend does not hold on the testing dataset, as the training dataset is noisy and the model did not learn relevant features that are useful for classifying the testing samples.

### 4.2.1 ANALYSIS OF FEATURE MAP

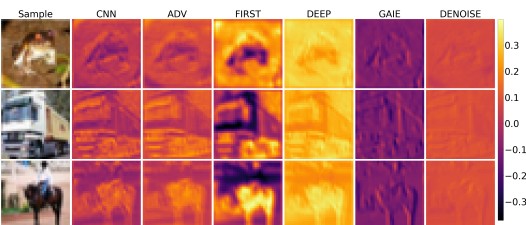

Figure 3: Feature maps from the first convolutional layer of CNNs trained with different strategies. From left to right: original input (first column), standard backpropagation, PGD adversarial training, GFA in the first layer, GFA in the first three layers, GAIE regularization, and feature denoising strategy.

Figure 3 illustrates the feature maps extracted from the first convolutional layer of CNNs trained with different strategies. Using the model trained by standard backpropagation (**CNN**) as a benchmark, we observe that the feature map produced by the adversarially trained model (**ADV**) is slightly shifted toward the positive activation range. The feature map of the model trained with GFA in the first layer (**FIRST**) shows increased contrast, highlighting sharper feature representations. When the alignment regularization is extended to the first three layers (**DEEP**), the feature map shifts significantly further in the positive direction, indicating enhanced and consistent activation patterns. In contrast, the model trained with GAIE regularization (**GAIE**) exhibits a notable negative shift in the feature map distribution, suggesting a different internal representation bias, likely due to its edge-focused nature. The model trained with feature denoising (**DENOISE**) produces subtle contrast feature maps, indicating suppression of both signal and noise, as mentioned in (Xie et al., 2019a).

These observations support our hypothesis that GAIE primarily captures edge features, while meaningful object representations often require more than edge information—such as color, contrast, texture and sharpness. Therefore, GFA offers a more comprehensive and robust learning signal. In fact, **GAIE** can be viewed as a special case of our GFA strategy, focusing narrowly on object edges. The enhanced robustness of the **DEEP** model over **FIRST** may be attributed to its ability to block and filter adversarial noise across multiple layers, particularly in deeper layers. This layer-wise suppression of perturbations enables the **DEEP** model to maintain more stable and meaningful feature representations under attack. Experimental results indicate that GFA applied to the first layer is most effective in mitigating adversarial perturbations, whereas its application in deeper layers offers diminished, albeit non-negligible, robustness benefits.

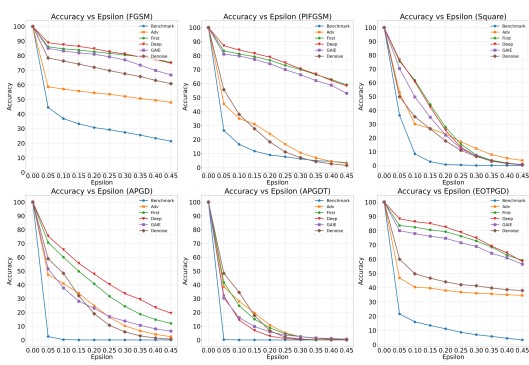

Figure 4: Evaluation of model accuracy degradation under increasing adversarial perturbation strength ($\epsilon$) for CNNs trained with different defense strategies against FGSM, PIFGSM, Square, APGD, APGDT, and EOTPGD attacks.

### 4.2.2 COMPARATIVE ANALYSIS OF MODEL ROBUSTNESS AGAINST GRADIENT-BASED ATTACKS ACROSS DIFFERENT TRAINING STRATEGIES

All training strategies are repeated five times, and the average performance for each strategy is reported in Table 2. We observe that the model trained with the **DEEP** strategy achieves the highest robustness across most attackers (attackers are configured with identical parameters). An exception occurs with the **APGDT** attacker, where the model trained using the **DENOISE** strategy demonstrates the best robustness. Additionally, the **FAB** attacker appears to be relatively ineffective under the current configuration, offering limited degradation across all models.

Table 2: Performance of CNN Models Trained with Different Strategies Against Various Adversarial Attacks on CIFAR-10

| Attacker[1] \ Strategy | Benchmark | PGD ADV | First | Deep | GAIE | Denoise |
|---|---|---|---|---|---|---|
| APGD[2] | $0.12 \pm 0.07$ | $42.52 \pm 1.2$ | $55.41 \pm 3.59$ | $\mathbf{57.82} \pm 5.42$ | $33.79 \pm 3.38$ | $43.03 \pm 3.29$ |
| APGDT[2] | $0.0 \pm 0.0$ | $28.04 \pm 1.4$ | $21.53 \pm 4.94$ | $19.35 \pm 6.93$ | $15.42 \pm 2.66$ | $\mathbf{29.34} \pm 2.91$ |
| FAB[2] | $98.03 \pm 0.22$ | $97.1 \pm 0.23$ | $\mathbf{98.55} \pm 0.22$ | $98.51 \pm 0.37$ | $97.91 \pm 0.47$ | $96.68 \pm 0.39$ |
| Square[2] | $8.13 \pm 0.51$ | $27.9 \pm 4.04$ | $54.34 \pm 4.5$ | $\mathbf{57.25} \pm 3.27$ | $45.13 \pm 2.4$ | $28.2 \pm 4.87$ |
| FGSM | $27.42 \pm 1.64$ | $53.17 \pm 2.45$ | $77.21 \pm 1.81$ | $\mathbf{77.86} \pm 2.36$ | $70.26 \pm 4.18$ | $62.57 \pm 3.48$ |
| MIFGSM | $12.48 \pm 0.98$ | $23.68 \pm 1.5$ | $74.98 \pm 2.03$ | $\mathbf{76.74} \pm 3.55$ | $63.56 \pm 5.19$ | $17.15 \pm 2.83$ |
| NIFGSM | $10.87 \pm 1.35$ | $24.48 \pm 1.25$ | $67.7 \pm 5.83$ | $\mathbf{73.35} \pm 3.68$ | $56.22 \pm 11.05$ | $17.47 \pm 2.75$ |
| DIFGSM | $10.03 \pm 2.48$ | $33.47 \pm 1.56$ | $54.71 \pm 3.99$ | $\mathbf{56.41} \pm 4.11$ | $42.37 \pm 4.46$ | $30.44 \pm 2.64$ |
| FFGSM | $24.52 \pm 1.6$ | $43.56 \pm 1.75$ | $77.54 \pm 1.64$ | $\mathbf{79.37} \pm 1.86$ | $73.67 \pm 2.57$ | $48.53 \pm 3.28$ |
| PIFGSM | $11.15 \pm 1.59$ | $24.8 \pm 1.75$ | $71.47 \pm 2.63$ | $\mathbf{73.71} \pm 3.49$ | $68.18 \pm 4.93$ | $15.34 \pm 2.66$ |
| PIFGSMPP | $8.93 \pm 1.69$ | $21.63 \pm 1.78$ | $71.84 \pm 2.53$ | $\mathbf{74.31} \pm 3.8$ | $64.43 \pm 5.2$ | $13.48 \pm 2.57$ |
| BIM | $12.62 \pm 1.24$ | $24.15 \pm 1.9$ | $75.07 \pm 2.49$ | $\mathbf{77.01} \pm 3.48$ | $64.72 \pm 5.12$ | $16.29 \pm 2.67$ |
| EOTPGD | $12.29 \pm 0.7$ | $39.78 \pm 1.75$ | $75.34 \pm 2.66$ | $\mathbf{77.82} \pm 2.6$ | $68.55 \pm 3.47$ | $39.54 \pm 3.21$ |
| PGD | $8.28 \pm 1.13$ | $24.62 \pm 2.02$ | $73.17 \pm 2.57$ | $\mathbf{74.57} \pm 3.6$ | $62.44 \pm 4.69$ | $16.55 \pm 2.73$ |
| PGDL2 | $18.78 \pm 0.95$ | $24.96 \pm 2.39$ | $80.44 \pm 1.46$ | $\mathbf{84.16} \pm 1.48$ | $71.73 \pm 3.44$ | $27.63 \pm 3.28$ |

1 All adversarial attacks are executed under identical configuration settings across models. Consequently, the attack intensity remains consistent for all gradient-based methods. Therefore, individual configuration details are omitted from this table for brevity.

2 The adversarial robustness of the CNN model is evaluated using the four components of the AutoAttack suite: APGD, APGDT, Square, and FAB. In this study, we report the results of each individual attacker separately, rather than using the combined AutoAttack pipeline (Croce & Hein, 2020).

Overall, the model trained with the **DEEP** strategy consistently exhibits stronger robustness than the one trained with the **FIRST** strategy, with an improvement of approximately 1–3% in accuracy across most attackers. The **FIRST** model, in turn, outperforms the **GAIE**, **ADV**, and **DENOISE** models, achieving roughly 5–15% higher accuracy compared to GAIE under the majority of attack configurations. These results underscore the effectiveness of applying GFA across multiple layers in suppressing adversarial perturbations and enhancing model robustness.

Figure 4 complements Table 2 by illustrating how model robustness evolves as the perturbation magnitude $\epsilon$ increases. The trends shown in Figure 4 are generally consistent with the results in Table 2. Notably, for the **APGDT** attacker, the model trained with the **DENOISE** strategy continues to outperform all other strategies when the $\epsilon$ less than 0.15. In contrast, under the **Square** attack, both the **FIRST** and **DEEP** models exhibit similar performance as $\epsilon$ increases, indicating that deeper regularization does not offer a significant advantage in this case. For the FGSM-based attackers, including FGSM and PIFGSM, models trained with **DEEP**, **FIRST**, and **GAIE** regularization consistently outperform those trained with **ADV** and **DENOISE**, especially as $\epsilon$ increases. These results further highlight the effectiveness of GFA in improving model robustness against a variety of attack strengths.

### 4.2.3 COMPARATIVE ANALYSIS OF MODEL ROBUSTNESS AGAINST OTHER ATTACKS ACROSS DIFFERENT TRAINING STRATEGIES

In Figure 5, the images (a) Deep Sparse Fool, (b) Jitter, and (c) EADEN attackers illustrate adversarial examples generated by attackers that do not rely on gradient-based methods but instead use optimization-based approaches. Taking the DeepFool attacker as an example, it aims to minimize the norm of perturbation. Therefore, DeepFool formulates the attacker as a constraint optimization problem, as shown in Equation 7.

$$f(x + r^*) \neq f(x) \; s.t. \; \|r^*\|_p \text{ is minimized} \tag{7}$$

where $\| \cdot \|_p$ represents the $L_p$-norm of the perturbation. At each iteration, the classifier is localized as a linear function around the current input $x_t$. If we assume that the classifier is differentiable, for each class $i$, the first-order Taylor expansion around $x_t$ can be obtained, as shown in Equation 8.

$$f_i(x) \approx f_i(x_t) + \nabla f_i(x_t)^T (x - x_t) \tag{8}$$

The decision boundary between the current class $k$ (where $k = \arg\max_i f_i(x)$ and any other class $j$ is approximated by a hyperplane. However, in the context of model trained by GFA regularization, the direction the DeepFool searches corresponds to radial directions of the input sample, which are the robustest direction naturally for most of the models.

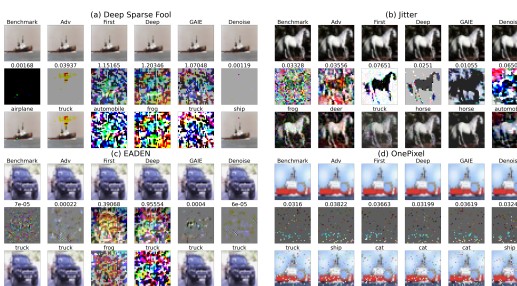

Figure 5: Comparison of the robustness of six training strategies (Benchmark, Adv, First, Deep, GAIE, Denoise) under four adversarial attacks: (a) Deep Sparse Fool, (b) Jitter, (c) EADEN, and (d) OnePixel. Each sub-image illustrates how a single sample is perturbed by one attacker and evaluated across different models. The numbers on top of the second row of each sub-figure indicate the noise intensity (measured in mean square error) applied to the perturbed samples.

As a result, the generated adversarial perturbations are no longer imperceptible, as shown in Figure 5. Advanced methods like C&W, EADEN, Jitter, and JSMA refine DeepFool's framework but fundamentally still explore radial directions in the input space. However, for most models, the radial direction is naturally the most robust. Consequently, these methods tend to generate large and visible perturbations, making their adversarial examples less effective, as seen in the examples from (a) Deep Sparse Fool, (b) Jitter, and (c) EADEN in Figure 5. More interestingly, for (b) Jitter, the generated adversarial noise simply takes the shape of a horse for the models trained with the GFA regularizer, since the background is consistently black. For (d) OnePixel attacker, a black-box and unstructured perturbation strategy, the perturbation it generates should be similar in size. It is also observed that the model trained with the Deep-Defense strategy has more robust performance on average. The consistent performance gains under a wide range of structured attacks highlight the effectiveness and general resilience of the proposed defense. Additional experimental results are provided in Appendix A.

## 5 CONCLUSION AND FUTURE WORK

This paper proposed *DeepDefense*, a simple yet powerful defense strategy that applies GFA across different layers of a neural network. By aligning the gradient of the loss with the input features at each layer, our method forces adversarial perturbations to move in directions that are naturally less effective. This results in a flatter and more stable loss landscape, helping models resist a wide range of attacks.

We supported our approach with both theoretical analysis and empirical experiments using CNNs and MLPs. The results showed that DeepDefense outperforms standard training, adversarial training, and other regularization methods in terms of robustness. In particular, DeepDefense-trained models require stronger and more visible perturbations to be fooled, which makes them more reliable and secure.

Notably, DeepDefense is model-agnostic, lightweight in implementation, and broadly compatible with standard architectures. These qualities position it as a practical and scalable solution for building more resilient deep learning systems. In future work, we plan to apply DeepDefense to larger models and more complex datasets. We are also interested in combining our method with other certified defenses to provide stronger guarantees of robustness. Finally, we will explore whether GFA can also improve the model's ability to understand important features in the input and adapt to new environments or slightly different data distributions.

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

TABLE OF CONTENTS

## A    MORE EXPERIMENT DATA FOR CNN MODEL TRAINED ON CIFAR10 DATASET

Through extensive experimentation and empirical observation, we find that incorporating gradient norm regularization at each layer can lead to a marginal improvement in model performance when training with the DeepDefense framework, especially when gradient-feature regularization is already applied. However, to better isolate and highlight the contribution of gradient-feature regularization, we focus solely on this technique in the current study. The combination of gradient norm regularization and gradient-feature regularization is identified as a promising direction and is therefore left for future exploration. We would also observed that the model trained with the gradient norm regularization only could not surpass the resistance of the models trained by the GFA, therefore, we do not discuss the models trained by GNR in the following experiments. It is observed that the GNR value is supposed to be small if the model is converged perfectly. Therefore, it is meaningless to discuss the GNR value solely.

### A.1    CNN ARCHITECTURE DESCRIPTION

The CNN model used in our experiments is a shallow convolutional network designed for CIFAR-10 classification. It consists of two convolutional layers, each followed by a max-pooling layer, and a fully connected head composed of three linear layers, as shown in Table 3.

Table 3: CNN Architecture Summary

| Layer Type | Input Shape | Output Shape | Param # |
|---|---|---|---|
| Conv2d | [Batch, 3, 32, 32] | [Batch, 32, 32, 32] | 896 |
| MaxPool | [Batch, 32, 32, 32] | [Batch, 32, 16, 16] | 0 |
| Conv2d | [Batch, 32, 16, 16] | [Batch, 32, 16, 16] | 9248 |
| MaxPool | [Batch, 32, 16, 16] | [Batch, 32, 8, 8] | 0 |
| Linear | [Batch, 2048] | [Batch, 1000] | 2048000 |
| Linear | [Batch, 1000] | [Batch, 600] | 600000 |
| Linear | [Batch, 600] | [Batch, 10] | 6000 |

## A.2 DEEPDEFENSE PERFORMANCE AGAINST GRADIENT-BASED ADVERSARIAL ATTACKS

FGSM is a one-step attack that generates adversarial examples by perturbing an input in the direction of the gradient of the loss function to maximize misclassification. The adversarial sample is computed using $x' = x + \epsilon \cdot \text{sign}(\nabla_x J(x, y))$, where $\epsilon$ controls the magnitude of the perturbation, $J(x, y)$ is the loss function, and $\nabla_x J(x, y)$ is its gradient with respect to the input (Goodfellow et al., 2014). Other FGSM-based attacks include the Momentum Iterative Fast Gradient Sign Method (MIFGSM) (Dong et al., 2018), Nesterov Iterative Fast Gradient Sign Method (NIFGSM) (Lin et al., 2019), Diverse Input Fast Gradient Sign Method (DIFGSM) (Xie et al., 2019b), Fast FGSM (FFGSM) (Wong et al., 2020), Projected Iterative Fast Gradient Sign Method (PIFGSM) (Gao et al., 2020a), and Projected Iterative Fast Gradient Sign Method++ (PIFGSM++) (Gao et al., 2020b), among others.

The Projected Gradient Descent (PGD) attack is an iterative, stronger variant of FGSM that generates adversarial examples by applying multiple small perturbation steps while ensuring the perturbed input remains within a predefined $l_p$-norm ball around the original input, as shown in Equation 9 (Mądry et al., 2017).

$$x_{t+1} = \Pi_{B_\epsilon(x)} \left( x_t + \alpha \cdot \text{sign}(\nabla_x J(x_t, y)) \right) \tag{9}$$

where $\Pi_{B_\epsilon(x)}$ projects $x_t$ back into the $l_p$-norm ball of radius $\epsilon$ around $x$, $\alpha$ is the step size, and $\nabla_x J(x_t, y)$ is the input gradient. Other PGD variants include the BIM (Kariyappa & Qureshi, 2019), Expectation Over Transformation Projected Gradient Descent (EOTPGD) (Liu et al., 2018c), and Projected Gradient Descent with $l_2$-Norm Constraint (PGDL2) (Mądry et al., 2017).

Figure 6 presents extended robustness evaluations of CNN models trained with different defense strategies under nine additional gradient-based attacks: DIFGSM, PIFGSMPP, FFGSM, MIFGSM, NIFGSM, BIM, FAB, PGD, and PGDL2. Each curve illustrates how model accuracy degrades as the adversarial perturbation strength ($\epsilon$) increases. These results, which complement the findings in Figure 4, further demonstrate the enhanced robustness of GFA-trained models—particularly the **DEEP** strategy. Notably, the DEEP strategy consistently outperforms the other approaches by a substantial margin across a wide range of attacks.

## A.3 DEEPDEFENSE PERFORMANCE AGAINST OPTIMIZATION-BASED ADVERSARIAL ATTACKS

### A.3.1 DETAILED ANALYSIS OF DEEPDEFENSE AGAINST DEEPFOOL ATTACK

For the DeepFool attacker algorithm, its aim is to minimize the norm of perturbation. Therefore, DeepFool formulates the attacker as a constraint optimization problem.

Given a classifier f: $\mathbb{R}^d \to \mathbb{R}^C$, where d is the input dimension and C is the number of classes, we aim to find the minimal perturbation $r^*$ such that the classifier assigns a different label to the perturbed input, as shown in Equation 10.

$$f(x + r^*) \neq f(x) \; s.t. \; \|r^*\|_p \text{ is minimized} \tag{10}$$

where $\| \cdot \|_p$ represents the $L_p$-norm of the perturbation.

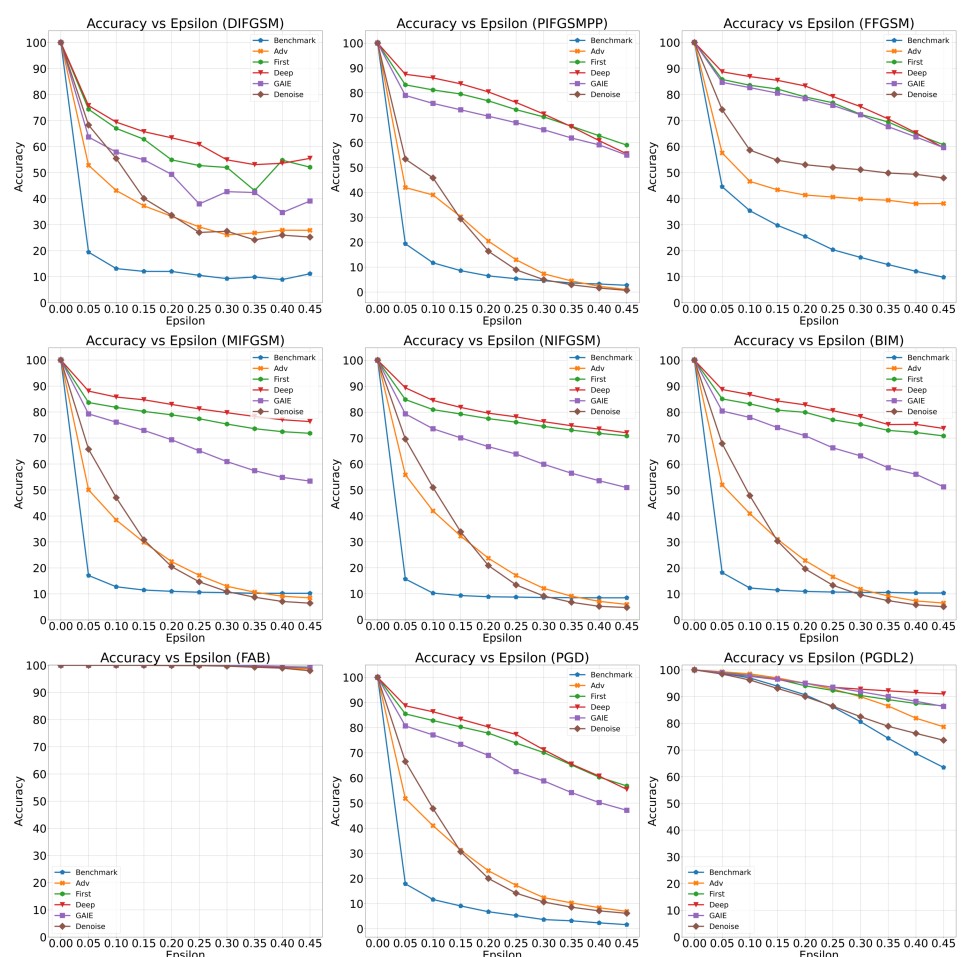

Figure 6: Accuracy curves of CNN models trained with different strategies under increasing adversarial perturbation strength ($\epsilon$) for additional gradient-based attacks, including DIFGSM, PIFGSMPP, FFGSM, MIFGSM, NIFGSM, BIM, FAB, PGD, and PGDL2.

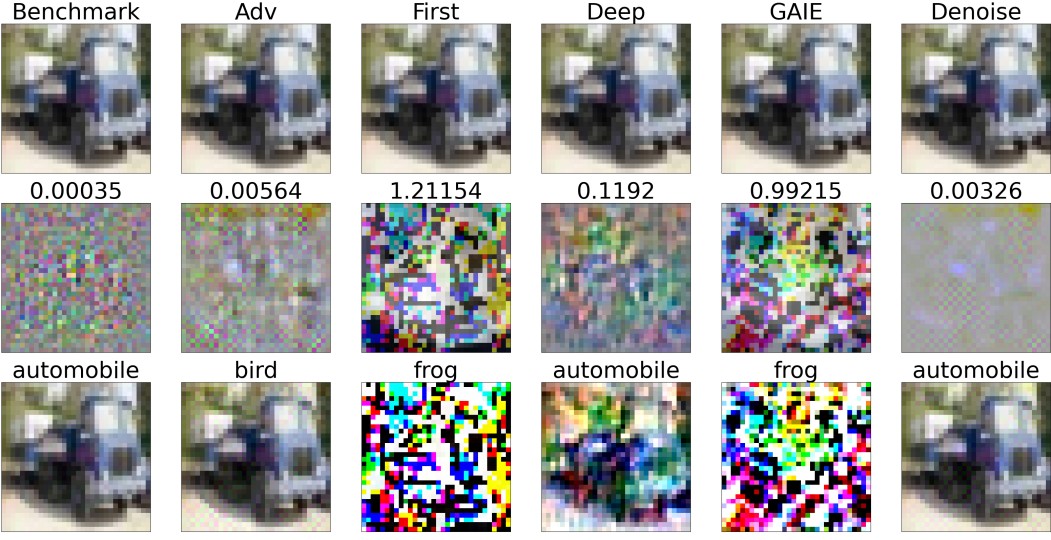

Figure 7: Models Attacked by Deep Fool Attacker. The numbers on top of the second row indicate the noise intensity applied to the perturbed samples.

At each iteration, the classifier is localized as a linear function around the current sample $x_t$. If we assume that the classifier is differentiable, for each class $i$, the first-order Taylor expansion around $x_t$ can be obtained, as shown in Equation 11.

$$f_i(x) \approx f_i(x_t) + \nabla f_i(x_t)^T (x - x_t) \tag{11}$$

The decision boundary between the current class $k$ (where $k = \arg\max_i f_i(x)$ and any other class $j$ is approximated by a hyperplane.

However, in deep defense, it aims to align the gradient with the feature, which means that the second term in the first-order Taylor expansion $\nabla f_i(x_t)^T (x - x_t)$ will be enlarged while the classifier can still return the correct labels. It counters to the DeepFool algorithm, nullifying its attack effect. As shown in Figure 7, if the deep fool wants to fool the model, the perturbation is very large and it is straightforward to be perceived by the person or detected by other approaches.

### A.3.2 Detailed Analysis of DeepDefense Against EADENL1 Attack

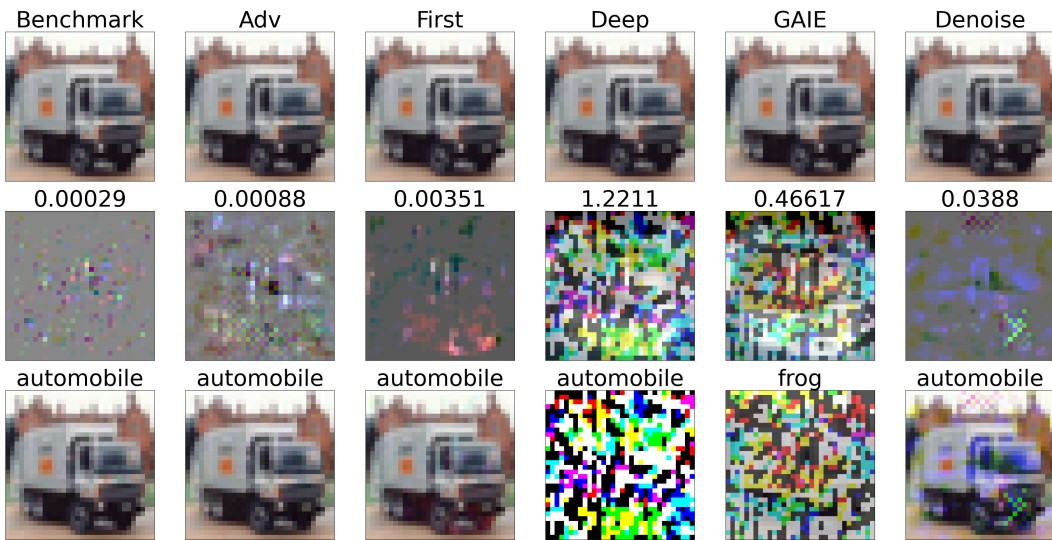

Figure 8: Models Attacked by EADENL1 Attacker. The numbers on top of the second row indicate the noise intensity applied to the perturbed samples.

EADEN extends the Carlini & Wagner (C&W) attack by incorporating Elastic-Net regularization, combining both $L_1$ and $L_2$ norms for enhanced sparsity and transferability. The general objective function for the EADEN is shown in Equation 12.

$$\min_x c \cdot f(x, t) + \beta \|x - x_0\|_1 + \|x - x_0\|_2^2, \ s.t. x \in [0, 1]^p \tag{12}$$

where $f(x, t) = \max\left(\max_{j \neq t} Z(x)_j - Z(x)_t, -\kappa\right)$ is the margin-based adversarial loss, and $\beta$ controls the sparsity of L1.

Then from the objective function, the gradient of input $x^k$ against the objective could be computed, as shown in Equation 13.

$$\nabla g(x^k) = c \cdot \nabla_x f(x^k, t) + 2(x^k - x_0) \tag{13}$$

where $\nabla_x f(x^k, t) = \nabla_x Z(x)_{j^*} - \nabla_x Z(x)_t$ with $j^* = \arg\max_{j \neq t} Z(x)_j$.

Then we can update the adversarial sample with Iterative Shrinkage-Thresholding Algorithm (ISTA), as shown in Equation 14.

$$x^{k+1} = S_\beta \left( x^k - \alpha_k \nabla g(x^k) \right) \tag{14}$$

where $S_\beta$ is the shrinkage-thresholding function, as shown in Equation 15.

$$S_\beta(z)_i = \begin{cases} \min(z_i - \beta, 1) & \text{if } z_i - x_{0,i} > \beta \\ x_{0,i} & \text{if } |z_i - x_{0,i}| \leq \beta \\ \max(z_i + \beta, 0) & \text{if } z_i - x_{0,i} < -\beta \end{cases} \tag{15}$$

It is observed that when $\beta = 0$, EADEN reduces to C&W ($L_2$-attack). The combination of $L_1$ and $L_2$ generates a sparse and imperceptible perturbations.

Now, assuming that the MSE is used as the loss function, then the input gradient against the loss function $\nabla_x J(x)$ and the $\nabla_x Z(x)$ is differ in one terms, as shown in Equation 16.

$$\nabla_x J(x) = \nabla_x Z(x) \cdot (Z(x) - y) \tag{16}$$

It is straightforward to note that for the rate of change of the i-th logit with respect to the j-th input feature is shown in Equation 17.

$$\nabla_x Z(x) = \gamma x \tag{17}$$

where $\gamma$ is a constant. Although, EADEN, EADL1, and C&W are all optimization-based and use iterations to generate the adversarial samples, it is highly possible that the perturbation direction is aligned with the direction of original samples, as shown in Figure 8, in which, the perturbation for the model trained by deepdefense achieve the best performace in defeating the adversarial samples generated by C&W, EADEN, and EADL1. It can easily observed that the perturbation direction is basically aligned with the original samples, making the adversarial attack less effective.

### A.3.3 DETAILED ANALYSIS OF DEEPDEFENSE AGAINST JSMA ATTACK

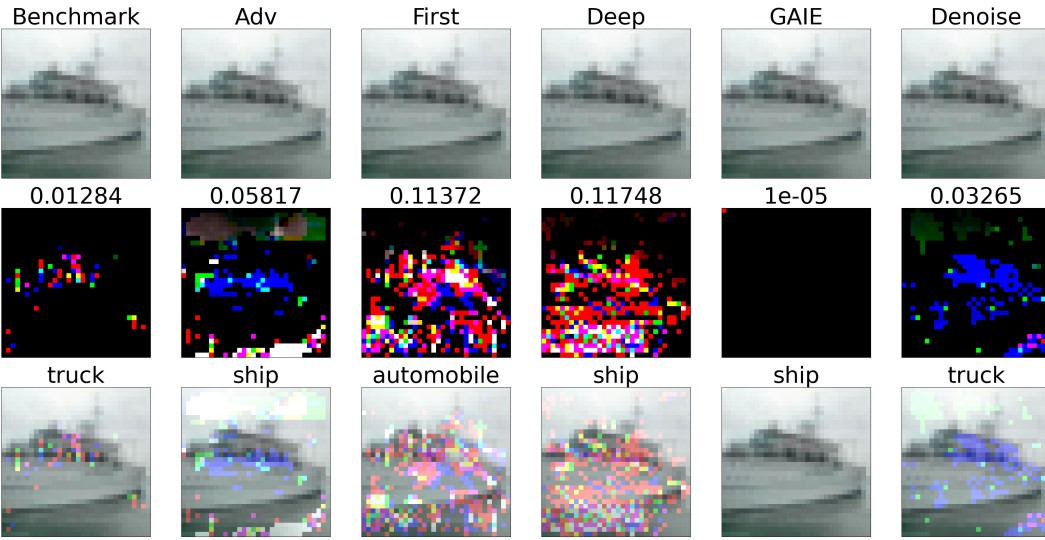

Figure 9: Models Attacked by JSMA Attacker. The numbers on top of the second row indicate the noise intensity applied to the perturbed samples.

The Jacobian-based Saliency Map Attack (JSMA) is a white-box, targeted adversarial attack that selectively modifies a small number of input features (pixels) to force a neural network into misclassifications. The objective function for the JSMA is similar to the DeepFool, as shown in Equation 18

$$f(x + r^*) \neq f(x) \ s.t. \ \|r^*\|_0 \text{ is minimized} \tag{18}$$

where $\| \cdot \|_0$ represents the $L_0$-norm of the perturbation.

Similar to C&W, EADEN, and EADL1, JSMA relies on the Jacobian matrix of the network's logits $Z(x)$, as shown in Equation 19.

$$J_f(x) = \left[ \frac{\partial Z_i(x)}{\partial x_j} \right]_{i,j} \tag{19}$$

where each entry represents the rate of change of the $i$-th logit with respect to the $j$-th input feature.

For a given target class $t$, we extract the input gradient, as shown in Equation 20.

$$\nabla_x Z_t(x) = \left[ \frac{\partial Z_t(x)}{\partial x_1}, \dots, \frac{\partial Z_t(x)}{\partial x_N} \right] \tag{20}$$

which tells us which features increase the probability of class $t$.

The saliency score $S(x)$ for each input feature $x_j$ is defined as in Equation 21.

$$S(x_j) = \frac{\partial Z_t(x)}{\partial x_j} - \sum_{k \neq t} \frac{\partial Z_k(x)}{\partial x_j} \tag{21}$$

which identifies features that increase $Z_t(x)$ (the logit of the target class) and decrease $Z_k(x)$ for all other classes $k \neq t$.

Once the Jacobian matrix $J_f(x)$ is computed, the feature $j^*$ with the highest saliency score was selected, as shown in Equation 22.

$$j^* = \arg\max_j S(x_j) \tag{22}$$

Then we could modify the most salient features to fool the model misclassify the images, as shown in Equation 23.

$$x_{j^*}^{(t+1)} = x_{j^*}^{(t)} + \alpha \cdot \text{sign}\left( \frac{\partial Z_t(x)}{\partial x_{j^*}} \right) \tag{23}$$

While, similarly to the analysis for C&W, EADEN, and EADL1, the each term in the Jacobian matrix is only different from the input gradient in one term, and they are all aligned with the original samples, but differ in intensity. Therefore, the perturbation generated by the JSMA based on the model trained by deepdefense strategy, even if it is a sparse attacker, the pixels that need to be adjusted is more evenly distributed across the entire images, as shown in Figure 9, in which the model trained by deep defense (first, consistency and alternation) strategy, the updated pixels are more evenly distributed than the other methods.

# B  ADDITIONAL EXPERIMENTS FOR MLP MODEL TRAINED ON FASHION MNIST

## B.1  ARCHITECTURE OF MLPS IN THE EXPLORATION

To further evaluate the effectiveness and generalizability of the proposed DeepDefense framework, we conduct additional experiments using a Multilayer Perceptron (MLP) trained on the Fashion MNIST dataset. The architecture of the MLP model used in this study is summarized in Table 4. It consists of three fully connected layers: the first layer maps the 784-dimensional flattened input image to 600 neurons, the second layer reduces it to 300 neurons, and the final output layer consists of 10 neurons corresponding to the 10 fashion categories. The hyperbolic tangent (tanh) function is used as the activation function for the hidden layers.

We deliberately use a three-layer structure to enable the evaluation of DeepDefense across different depths of the MLP. This design allows us to analyze how applying Gradient-Feature Alignment (GFA) regularization at different layers—individually or jointly—impacts model robustness against adversarial attacks. By comparing strategies such as applying GFA to the first layer only versus applying it to the first two layers (DEEP), we aim to understand how the depth of regularization affects adversarial resilience in fully connected networks.

Table 4: MLP Architecture Summary

| Layer Type | Input Shape | Output Shape | Param # |
|---|---|---|---|
| Linear | [Batch, 784] | [Batch, 600] | 470400 |
| Linear | [Batch, 600] | [Batch, 300] | 180000 |
| Linear | [Batch, 300] | [Batch, 10] | 3000 |

## B.2 LAYER-WISE GFA REGULARIZATION VALUES FOR MLPS

Table 5: GFA Regularization Values for Models Trained with Different Strategies[1]

| Strategies | Dataset | GFA Regularization Value | | |
|---|---|---|---|---|
| MLP | train | $0.0003 \pm 0.0013$ | $0.0002 \pm 0.0015$ | $0.0001 \pm 0.0028$ |
| | test | $0.0029 \pm 0.0013$ | $0.0019 \pm 0.0015$ | $0.0051 \pm 0.0024$ |
| ADV | train | $-0.0096 \pm 0.0048$ | $-0.0146 \pm 0.0043$ | $-0.0149 \pm 0.0082$ |
| | test | $-0.0022 \pm 0.0044$ | $-0.0115 \pm 0.0043$ | $-0.0116 \pm 0.0075$ |
| FIRST | train | $0.985 \pm 0.0012$ | $-0.013 \pm 0.001$ | $-0.0277 \pm 0.0472$ |
| | test | $0.8417 \pm 0.0049$ | $-0.0117 \pm 0.0009$ | $-0.0232 \pm 0.0442$ |
| DEEP | train | $0.9694 \pm 0.0017$ | $0.9376 \pm 0.0035$ | $0.7469 \pm 0.0463$ |
| | test | $0.85 \pm 0.005$ | $0.8468 \pm 0.0051$ | $0.6946 \pm 0.0439$ |

[1] Although GFA regularization is explicitly applied only in **FIRST** and **DEEP**, GFA values are calculated for all models to evaluate the impact of GFA on model robustness. For comparison purpose, on the training dataset, at the first layer, the GFA value difference between **FIRST** and **DEEP** is not significantly.

In our evaluation of layer-wise GFA regularization on MLPs trained with the FashionMNIST dataset, we observed that the GFA values at the first layer are quite similar between the FIRST and DEEP strategies on the training set. This design ensures a fair comparison by isolating the influence of deeper layer alignment. Notably, FashionMNIST, being a relatively clean and structured dataset, allows GFA regularization to generalize smoothly from the training to the testing distribution.

Interestingly, the MLP trained with adversarial training (ADV) exhibited slightly higher GFA values compared to the standard MLP, suggesting that GFA may capture underlying changes in the gradient landscape introduced by adversarial learning. This subtle increase implies a potential connection between GFA values and the geometry of the loss surface—specifically, that larger GFA values may correspond to flatter loss surfaces, which in turn are associated with improved robustness. This observation supports the hypothesis that GFA not only regularizes gradient direction but also serves as an implicit indicator of model sensitivity to adversarial noise.

## B.3 EVALUATION OF ADVERSARIAL ROBUSTNESS IN MLPS TRAINED WITH DIFFERENT STRATEGIES

Table 6 presents the adversarial robustness of MLPs trained with various strategies, assessed against a wide range of attacks. Overall, the DEEP strategy, which applies GFA regularization across multiple layers, consistently achieves the highest robustness across most attackers. This includes both gradient-based and optimization-based methods, demonstrating that deeper alignment of gradients with feature representations enhances the model's resilience to adversarial perturbations.

However, for a subset of attacks—namely APGD, APGDT, and Square—the model trained using the FIRST strategy (GFA applied only at the first layer) slightly outperforms the DEEP-trained model in

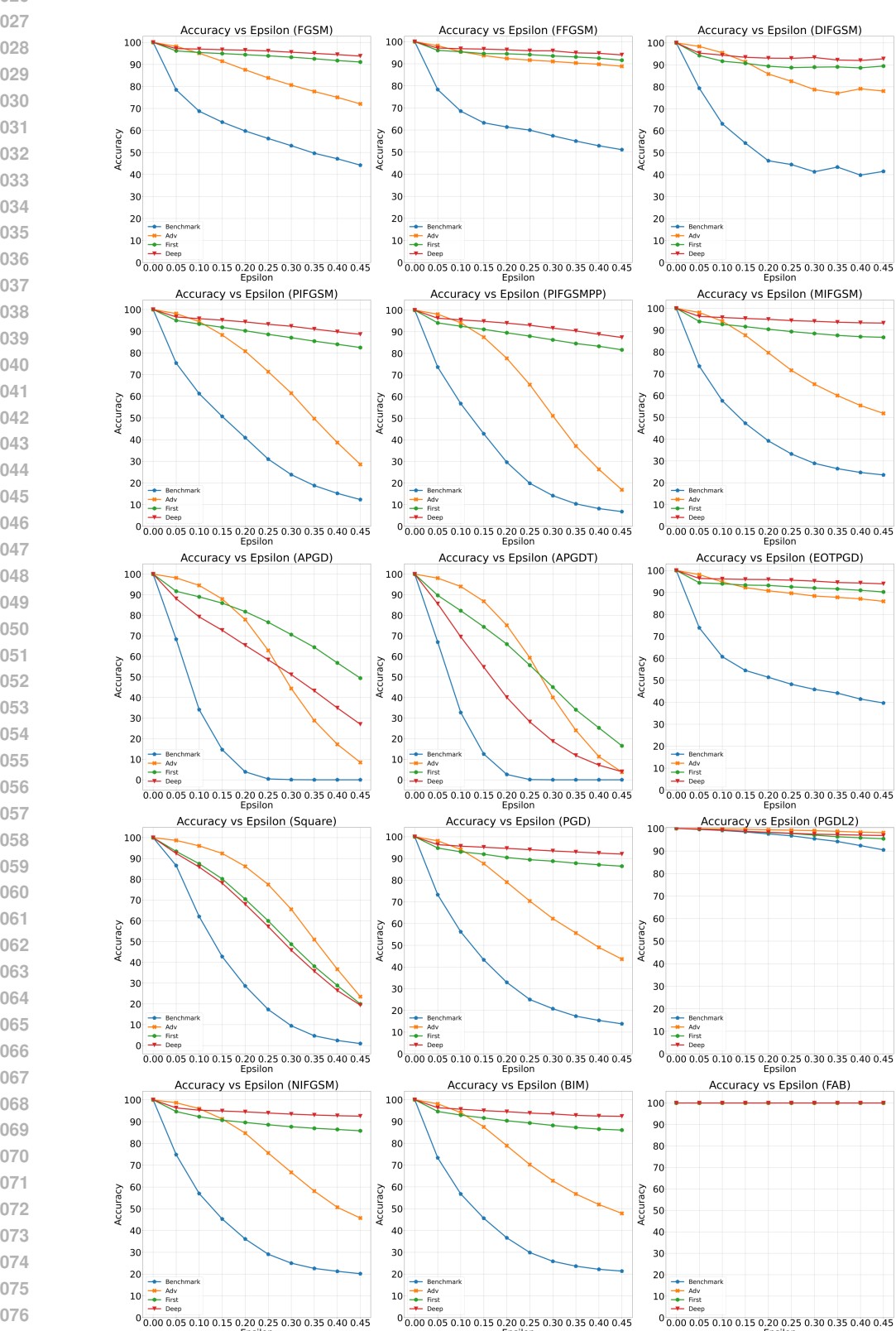

Figure 10: MLPs Attacked by Gradient-based Adversarial Attackers with Increasing $\epsilon$

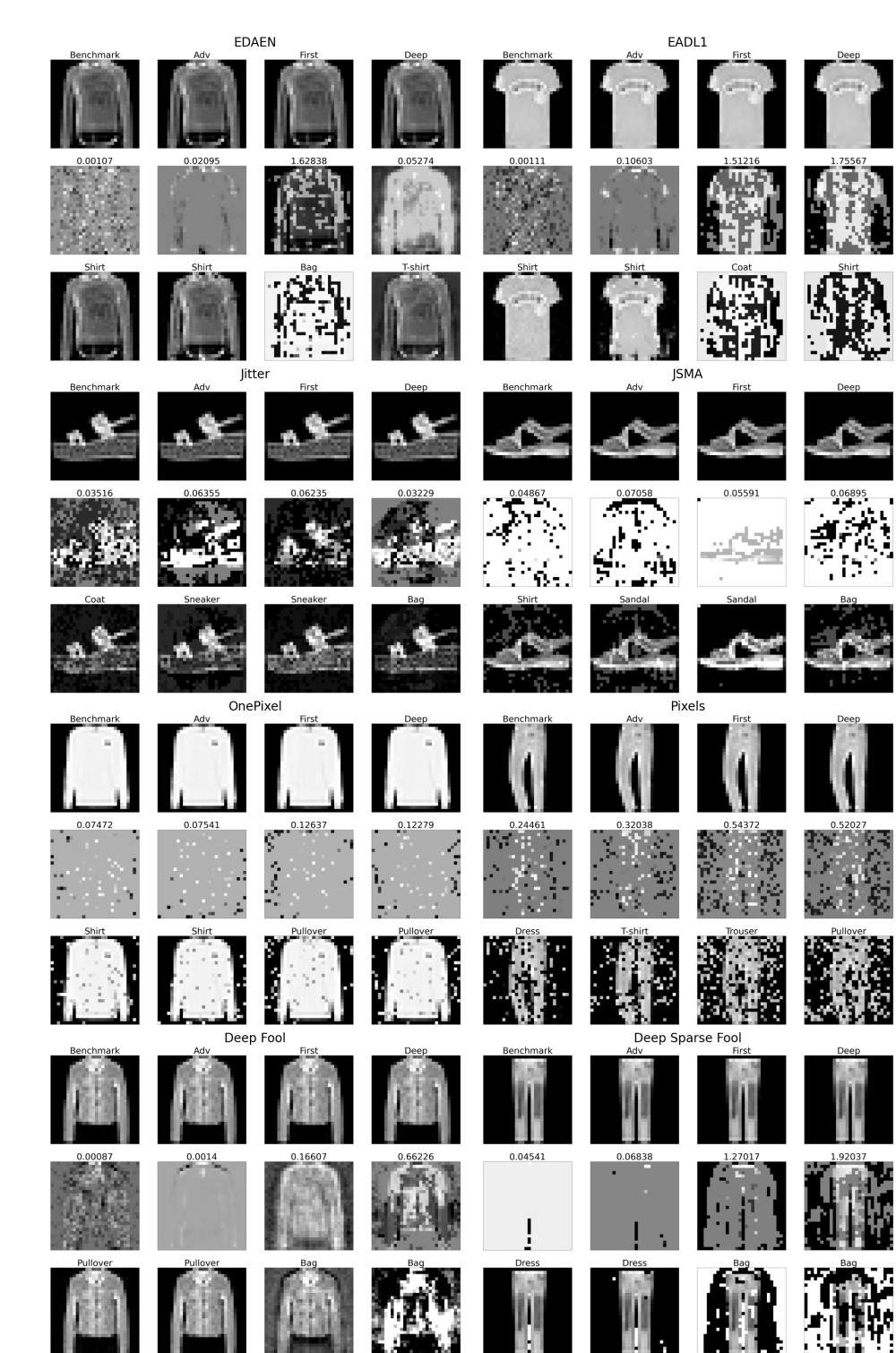

Figure 11: MLPs Attacked by Optimization-based Adversarial Attackers

Table 6: Performance of MLP Trained with Different Strategies Against Various Adversarial Attacks

| Attacker[1] \ Strategy | Benchmark | | PGD ADV | | First | | Deep | |
|---|---|---|---|---|---|---|---|---|
| | Intensive | Accuracy | Intensive | Accuracy | Intensive | Accuracy | Intensive | Accuracy |
| APGD | $0.0545 \pm 0.0008$ | $0.42 \pm 0.36$ | $0.0354 \pm 0.0011$ | $42.09 \pm 2.15$ | $0.0121 \pm 0.0024$ | **73.43** $\pm 2.6$ | $0.0265 \pm 0.0037$ | $43.46 \pm 6.7$ |
| APGDT | $0.0432 \pm 0.0004$ | $0.0 \pm 0.01$ | $0.036 \pm 0.001$ | $38.18 \pm 2.01$ | $0.0254 \pm 0.0005$ | **43.54** $\pm 2.3$ | $0.0385 \pm 0.0011$ | $16.96 \pm 2.19$ |
| Square | $0.0591 \pm 0.0006$ | $10.88 \pm 1.07$ | $0.0239 \pm 0.0006$ | $64.05 \pm 0.96$ | $0.0342 \pm 0.0022$ | **49.11** $\pm 2.8$ | $0.0354 \pm 0.0017$ | $47.85 \pm 2.61$ |
| FGSM | $0.1076 \pm 0.0001$ | $45.8 \pm 0.99$ | $0.1059 \pm 0.0005$ | $73.84 \pm 0.86$ | $0.0573 \pm 0.0002$ | $92.22 \pm 0.34$ | $0.0573 \pm 0.0003$ | **94.0** $\pm 0.39$ |
| MIFGSM | $0.0509 \pm 0.0002$ | $28.17 \pm 0.99$ | $0.0507 \pm 0.0003$ | $63.74 \pm 1.16$ | $0.0332 \pm 0.0001$ | $89.14 \pm 0.53$ | $0.0334 \pm 0.0001$ | **93.58** $\pm 0.38$ |
| NIFGSM | $0.0518 \pm 0.0002$ | $24.19 \pm 1.21$ | $0.0481 \pm 0.0001$ | $65.2 \pm 1.06$ | $0.0333 \pm 0.0001$ | $88.05 \pm 0.51$ | $0.0335 \pm 0.0001$ | **92.91** $\pm 0.31$ |
| DIFGSM | $0.03 \pm 0.0007$ | $40.36 \pm 1.36$ | $0.0332 \pm 0.0005$ | $77.03 \pm 0.99$ | $0.0176 \pm 0.0011$ | $89.45 \pm 0.44$ | $0.0164 \pm 0.0008$ | **91.35** $\pm 0.73$ |
| FFGSM | $0.0308 \pm 0.0$ | $55.56 \pm 0.94$ | $0.0304 \pm 0.0002$ | $90.29 \pm 0.55$ | $0.0181 \pm 0.0$ | $93.88 \pm 0.21$ | $0.0175 \pm 0.0001$ | **95.28** $\pm 0.21$ |
| PIFGSM | $0.061 \pm 0.0001$ | $22.54 \pm 1.27$ | $0.0613 \pm 0.0003$ | $59.11 \pm 1.7$ | $0.035 \pm 0.0002$ | $87.67 \pm 0.52$ | $0.0348 \pm 0.0001$ | **91.84** $\pm 0.39$ |
| PIFGSMPP | $0.0616 \pm 0.0002$ | $14.56 \pm 1.14$ | $0.0619 \pm 0.0002$ | $48.6 \pm 2.14$ | $0.035 \pm 0.0002$ | $86.86 \pm 0.58$ | $0.0348 \pm 0.0001$ | **91.12** $\pm 0.37$ |
| BIM | $0.0523 \pm 0.0005$ | $25.42 \pm 1.16$ | $0.0529 \pm 0.0002$ | $61.23 \pm 1.32$ | $0.0329 \pm 0.0001$ | $88.85 \pm 0.47$ | $0.0332 \pm 0.0001$ | **92.89** $\pm 0.35$ |
| EOTPGD | $0.0288 \pm 0.0$ | $44.9 \pm 0.96$ | $0.0293 \pm 0.0002$ | $87.92 \pm 0.49$ | $0.0174 \pm 0.0001$ | $92.48 \pm 0.26$ | $0.0169 \pm 0.0$ | **94.71** $\pm 0.16$ |
| PGD | $0.0526 \pm 0.0004$ | $20.3 \pm 1.48$ | $0.0536 \pm 0.0002$ | $60.65 \pm 1.69$ | $0.0316 \pm 0.0001$ | $89.07 \pm 0.5$ | $0.0318 \pm 0.0001$ | **93.06** $\pm 0.3$ |
| PGDL2 | $0.005 \pm 0.0$ | $49.94 \pm 1.07$ | $0.005 \pm 0.0$ | $70.06 \pm 0.28$ | $0.0049 \pm 0.0$ | $91.57 \pm 0.34$ | $0.0049 \pm 0.0$ | **94.98** $\pm 0.23$ |

[1] All the attackers are using the same configuration when attacking the model, the noise intensive is demonstrated to evaluate the attack intensity of the corresponding attacker.

terms of test accuracy. Despite this, a closer examination of the adversarial perturbation intensity reveals that the differences in robustness between FIRST and DEEP under these attacks are relatively minor. Both models exhibit comparable levels of resistance to perturbations, suggesting that the marginal advantage of FIRST in these cases may stem from stochastic variation or overfitting to early-layer representations, rather than a substantial difference in defense effectiveness.

Figure 10 illustrates the robustness of MLP models trained with different strategies as the adversarial perturbation strength ($\epsilon$) increases under various gradient-based attackers. It is evident that the MLP trained using the DEEP strategy consistently outperforms other training strategies across most attacks. The performance gap becomes especially pronounced as the perturbation magnitude grows, demonstrating the superior stability and resilience provided by layer-wise GFA regularization.

However, an exception is observed with APGD, APGDT, and Square attacks, where the model trained with gradient-feature alignment at the first layer (FIRST strategy) achieves slightly higher accuracy than the DEEP-trained model. This observation is consistent with the results reported in Table 6, confirming that under specific attack configurations, GFA applied solely at the first layer may offer marginal benefits.

Figure 11 presents the adversarial robustness of MLP models under various optimization-based attacks. It is evident that the MLPs trained using FIRST and DEEP strategies exhibit significantly higher resistance to perturbations compared to those trained using standard backpropagation or adversarial training (ADV). In particular, the noise intensity required to successfully mislead the FIRST and DEEP models is 10 to 100 times greater than that required to attack the standard MLP model. This substantial increase in required perturbation magnitude indicates that GFA regularization effectively enhances robustness by making the loss landscape flatter and reducing vulnerability to optimization-based attackers such as C&W, EADEN, and DeepFool. The DEEP model, which applies GFA across all layers, consistently requires the strongest perturbations, underscoring the advantage of layer-wise alignment in resisting such attacks.

The adversarial perturbations generated by Jitter and JSMA attacks appear to be comparable in magnitude across all MLP models, particularly for those trained with GFA strategies such as FIRST and DEEP. This similarity may stem from variations in sample difficulty—certain input samples are inherently more susceptible to misclassification than others, regardless of the attacker used. Consequently, even strong defense mechanisms may exhibit variability in robustness depending on the specific input. To better understand this phenomenon, further experiments are necessary, including per-sample robustness analysis and stratification by difficulty level. This would provide deeper insights into how GFA regularization interacts with different attack types and input complexities.

## STATEMENT ON AI WRITING ASSISTANCE

ChatGPT was used to improve grammar and refine sentence structure, with all AI-generated edits carefully reviewed and adjusted for relevance.

