# OpenReview forum: "DeepDefense: Layer-Wise Gradient-Feature Alignment for Building Robust Neural Networks"
_ICLR.cc/2026/Conference — ICLR 2026 Conference Withdrawn Submission_

### Official Review · Reviewer_w3sQ · 2025-10-25

**Soundness:** 2
**Presentation:** 2
**Contribution:** 2
**Rating:** 2
**Confidence:** 4

**Summary:**

- Proposes DeepDefense with a Gradient–Feature Alignment (GFA) regularizer: encourages the loss gradient to align (via cosine similarity) with inputs or intermediate features.

- Intuition: if gradients align with features, the loss changes mostly in the radial direction and is flatter along tangential directions, intended to make common adversarial perturbations less effective.

- Two variants: FIRST (apply GFA at the first layer) and DEEP (apply GFA at multiple layers)

**Strengths:**

- Simple, model-agnostic regularizer: GFA is easy to plug in (cosine penalty), with clear training recipe and minimal overhead.

- Compelling geometric intuition + diagnostics: Radial/tangential view is clear, supported by gradient/feature visualizations and layer-wise analysis.

- Thoughtful ablations & broad attack menu: FIRST vs DEEP comparisons, ε-sweeps, and many attack families provide a reasonably thorough study.

**Weaknesses:**

- Non-standard evaluation choices: Tests are run only on samples all models classify correctly, and AutoAttack components are reported separately instead of the canonical combined AA, both can inflate robustness and hinder comparability.

- Attack configs under-specified / possibly weak: Extremely high accuracies under FAB (and others) suggest misconfiguration; key details like ε, steps, and restarts aren’t consistently specified.

- Missing clean-accuracy trade-offs

- Narrow scope of models/data: Evidence is limited to a small CNN on CIFAR-10 and an MLP on Fashion-MNIST; no results on modern backbones (e.g., WRN/ResNet/ViT) or harder datasets.

- Baselines not state-of-the-art: Comparisons omit strong recent methods (e.g., tuned TRADES, PNI, Learn2Perturb), leaving the true competitiveness uncertain.

- Key citations missing (e.g. PGD, DeepFool, etc.)

- The idea of flatness is not very clear, and Figure 2 does not do a great job helping with - moreover, analysis and connection to other studies in the sharpness/flatness space (e.g SAM) is not provided

- although the paper mentions they provide theoretical evidence, essentially it's not there - it is barely suggested that the GFA causes flatness and there is no numerical or theoretical evidence for it - moreover, the connections between per layer flatness and adversarial robustness is not quite clear

- Figure 1 could be made more clear and elaborate

- Table 1 looks almost unnecessary - first of all the idea regularization values and its importance are vague and second, I cannot see any reason why that table is there.

**Questions:**

- Although I understand a potential lack of computational resources, I believe for a method either the theoretical side or the empirical side should be strong enough - so given that my suggestion for the authors, in case they want to focus on smaller models, is to provide deeper analysis of why is the alignment important, and what value does it provide - moreover, what is the effect on clean accuracy and regular models - how would this extend to bigger models and potentially LLMs/VLMs.

---

### Official Review · Reviewer_Sj1x · 2025-10-27

**Soundness:** 2
**Presentation:** 2
**Contribution:** 2
**Rating:** 2
**Confidence:** 5

**Summary:**

This paper proposes DeepDefense, a defense framework based on layer-wise Gradient-Feature Alignment (GFA) regularization to improve the robustness of deep neural networks against adversarial attacks. The method aligns input gradients with internal feature representations across layers to encourage a smoother loss landscape and reduce sensitivity to perturbations. The authors provide limited theoretical discussion and empirical evaluations on CIFAR-10 and Fashion-MNIST datasets using CNN and MLP architectures. Experimental results show that the proposed GFA regularization improves robustness over several baseline defenses on small-scale settings.

**Strengths:**

1. Simple and intuitive idea: The proposed Gradient-Feature Alignment regularization is conceptually straightforward and easy to implement.
2. Effective on small models and datasets: The experiments show that DeepDefense can enhance robustness for lightweight networks on small datasets.

**Weaknesses:**

1. Limited experimental scope:
    - The evaluation is restricted to outdated and small datasets (CIFAR-10, Fashion-MNIST). For a robustness paper, larger and more challenging datasets (e.g., CIFAR-100, Tiny-ImageNet) are necessary.
    - The network architectures (CNN, MLP) are too simple. The study should include modern architectures such as ResNet or WideResNet to demonstrate scalability.
    - The attack methods are mostly from 4–5 years ago. The paper lists the components of AutoAttack (APGD, FAB, Square, APGDT) separately rather than reporting the aggregated AutoAttack results, which is the standard practice.
    - The defense baselines are incomplete. More representative adversarial training methods and recent defense mechanisms should be compared.
2. Weak and superficial theoretical analysis: The so-called “theoretical analysis” mainly restates intuitive gradient alignment properties without any formal derivation or theorem. To make the theoretical contribution meaningful, the authors could analyze the inter-layer influence from a Lipschitz continuity perspective and provide formal results or bounds.
3. Poor mathematical writing and notation quality:
    - Equations are often presented as isolated blocks rather than integrated within the text.
    - Symbol definitions are inconsistent: $L$ and $\mathcal{L}$ are mixed; $x$ and $\mathbf{x}$ are used interchangeably; vector and scalar quantities are not clearly distinguished.
    - Norms are undefined, and dimensions of variables are often unclear. Overall, the mathematical presentation does not meet the expected standard for a top-tier venue.
4. Formatting and consistency issues:
    - The numerical precision in tables is inconsistent (e.g., varying decimal places).
    - Training time comparisons are missing, so it is unclear how computationally efficient DeepDefense is compared with adversarial training or other defenses.
    - Typos:
      - pseudoscope → pseudocode (page 5, line 220)
      - recursive → recursion (page 5, line 250)
      - impact → impacts (page 5, line 261)
      - performace → performance (page 17, line 888)

**Questions:**

See Weaknesses.

---

### Official Review · Reviewer_yyTY · 2025-10-29

**Soundness:** 3
**Presentation:** 2
**Contribution:** 2
**Rating:** 2
**Confidence:** 5

**Summary:**

This paper proposes DeepDefense, a defense framework designed to improve adversarial robustness by applying Gradient-Feature Alignment (GFA) regularization across multiple layers of a neural network. The central idea is to align input gradients with feature representations, thereby promoting smoother loss landscapes in tangential directions and reducing model sensitivity to adversarial perturbations. The authors argue that GFA suppresses loss variations along tangential directions where most attacks are effective. Empirically, DeepDefense is evaluated on CIFAR-10 using convolutional neural networks trained with layer-wise GFA regularization and tested against a wide set of gradient-based and optimization-based attacks.

**Strengths:**

The paper introduces an intuitive mechanism that connects gradient alignment and feature representation to local loss smoothness, providing geometric intuition for robustness.

The layer-wise GFA regularization is easy to integrate and lightweight, requiring only minor changes to standard training pipelines.

**Weaknesses:**

#### **Novelty concerns**
A central issue with this paper is its lack of novelty compared to prior gradient alignment defenses. In particular, the core principle of encouraging gradient alignment as a regularizer for adversarial training has already been extensively studied, most notably in *Understanding and Improving Fast Adversarial Training* (Andriushchenko & Flammarion, NeurIPS 2020), where the authors propose GradAlign, a method that maximizes the gradient alignment between inputs and their perturbed counterparts to prevent catastrophic overfitting and improve adversarial training. The methodology in DeepDefense appears conceptually and mathematically similar. However, this paper does not discuss GradAlign in sufficient detail in the related work, nor does it provide any empirical comparisons or arguments clarifying how GFA differs in terms of objective, behavior, or robustness.

Furthermore, related approaches such as *Adversarial Feature Alignment: Balancing Robustness and Accuracy in Deep Learning via Adversarial Training*(Park et al., AISEC 2024) also employ alignment-based defenses at the feature level. Despite methodological differences, these are neither included in the related work nor treated as empirical baselines.

#### **Gradient obfuscation and evaluation validity**
Another major concern is that the observed robustness improvements are likely due to gradient obfuscation rather than genuine robustness. The proposed regularization explicitly enforces alignment between gradients and features, flattening the loss landscape and reducing gradient magnitudes. While this superficially appears to improve robustness, it effectively hides useful gradient information from attackers, making standard gradient-based attacks artificially weaker. This issue has been extensively documented in the paper *Obfuscated Gradients Give a False Sense of Security* (Athalye et al., ICML 2018). As the main outcome of the paper from Athalye et al. (2018) is the need to test defenses against adaptive attackers to avoid a false sense of security caused by using weak attacks. The authors demonstrated that even minor updates to an attack can effectively counter gradient-obfuscation defenses. Since then, adaptive evaluation of defenses has become a de facto standard for evaluating robustness correctly (On Adaptive Attacks to Adversarial Example Defenses, NeurIPS 2020).
However, this paper does not perform adaptive attacks to test whether the defense truly offers stronger optimization objectives. To make it concrete, the authors should have conducted experiments with attackers with smarter objectives and capable of responding to the gradient obfuscation setup (e.g., using differences in the logits ratio loss, targeted attacks, or loss reparameterization).
Indeed, the presented results already hint at this issue: the defense fails most clearly against APGDT, which uses a different loss formulation and more adaptive optimization.

Overall, the current experimental setup lacks a fundamental evaluation scenario to conclude that the proposed approach offers realistic robustness improvements. The paper’s conclusion that GFA suppresses adversarial perturbations might instead reflect the fact that the gradients have become uninformative for specific attack implementations.

#### **Limited experimental coverage**
All experiments are conducted on a single CNN architecture using CIFAR-10, which does not offer a way to verify that GFA is model-agnostic and potentially adaptable to other architectures and tasks as claimed. In this regard,  the paper does not include evaluations on larger or more diverse models (e.g., ResNets, WideResNets, or Vision Transformers).

Furthermore, the experiments are limited to only CIFAR-10, discarding more challenging  and reliable datasets like ImageNet. Moreover, the paper fails to compare against standard and well-established adversarial defenses from RobustBench or other adversarial training schemas like TRAES, MART, or GradAlign itself.

Lastly, the evaluation runs AutoAttack components (APGD, FAB, Square) separately rather than as a unified benchmark.

#### **Presentation and structure**
The paper suffers from chaotic presentation. The layout makes extensive use of minipage formatting and extremely small figures, resulting in a visually dense, difficult-to-read layout. Figures 1–3 are really small, have inconsistent scaling, and are not colorblind-friendly.  Moreover, in Table 2, the label “BENCHMARK” is unclear as to what it refers to.

#### *Actionable Points*
As first and important aspect, I invite the authors to clarify novelty by explicitly contrasting DeepDefense with GradAlign and other alignment-based defenses. Provide a detailed explanation of the conceptual and methodological differences, and include direct empirical comparisons. As second important requirement, test against adaptive attacks to rule out gradient obfuscation. In this regard, the authors could evaluate the robustness considering  adaptive PGD integrated with different loss functions or strategies like the ones in Athalye et al., ICML 2018.
Furthermore, I invite the authors to expand the experimental coverage to include more architectures (ResNet, ViT) and stronger baselines (TRADES, MART, GradAlign, RobustBench models). Lastly, improve paper organization and figure design: avoid excessive minipage usage, enlarge figures, adopt colorblind-friendly palettes, and clearly define all labels (especially “BENCHMARK” in Table 2).

**Questions:**

How does GFA differ methodologically and conceptually from GradAlign (NeurIPS 2020)?

Is the robustness effect due to improved generalization or simply reduced gradient informativeness (gradient masking)?

Why do the authors use the MSE loss during training instead of the cross-entropy? The gradient directions from MSE and CE can differ substantially, which may affect the attacks and the gradient magnitude.

What is the accuracy of the trained model compared to the other defenses?

---

### Official Review · Reviewer_Lti5 · 2025-11-01

**Soundness:** 1
**Presentation:** 1
**Contribution:** 1
**Rating:** 0
**Confidence:** 5

**Summary:**

This paper presents DeepDefense, a straightforward defense strategy that implements GFA across various layers of a neural network. This method aligns the gradient of the loss with the input features at each layer, thereby directing adversarial perturbations to move in less effective directions.

**Strengths:**

I honestly don't see anything positive in this paper. I really do.

**Weaknesses:**

In the related work section (lines 136-137), the authors argue that the GNR has little effect on enhancing the robustness of neural networks.  I am curious about the source of this claim.  Are there any experiments that support this claim?

Also, in the related work section (lines 150-151), the authors argue that the GradAlign method, for instance, requires a robust neural network or a pre-trained teacher model. This claim is factually wrong. This statement is definitively false.  The GradAlign method is a fast adversarial training approach that enables simultaneous training of a classifier with both adversarial and clean samples from the start.

The presentation of Figure 2 is poor.  The resolution of this figure is unsuitable.  Additionally, the caption of this figure must include detailed information regarding the x-axis and y-axis, as well as the relationship between $(\alpha,\beta)$ in the figure and Eq.(2).

In lines 189-195, the authors should clearly differentiate between the previously mentioned intuition and the CURE [1] paper, if such a distinction exists!!

In lines 240-242 (subsection 3.2.1), there is a discrepancy in notation: the authors have consistently used "$L$" to denote layers of neural networks throughout the paper, yet here they switch to "$n$".

In line 292 of the paper, the authors claim that they rescaled all input samples to the range $\[-1,1\]$. In the robustness literature, it is well established that the threat model is $\[0,1\]$. I wondered, is there any reason for this choice? Whatever the reason, it is completely inconsistent with what we have seen from 2017 (CW paper) to today.

In line 309, the mentioned ADV method is the Madry defense [2] that should be cited clearly.

In lines 323-325, this observation is made: "Although ADV is not explicitly trained with GFA, its GFA on the training dataset is slightly higher than that of the model trained with standard backpropagation." This observation is one of the selling points of the CURE [1] paper.
The authors must establish a connection between their observation and observations in the CURE [1] paper.

In lines 399-402, Has anybody ever asked the authors why Auto-Attack has received more than 2000 citations?  Why would Auto-Attack be expected if we were to list robust accuracy individually according to those four attacks, considering they were already introduced?  Because those four attacks are mutually exclusive, we prioritize robust accuracy under Auto-Attack.

**Questions:**

Please see weakness section

**Details Of Ethics Concerns:**

In this paper, there are several papers and methods that must be cited but authors ignored them.

---

### Note · Authors · 2025-11-28

**Comment:**

Additional experiments are required to adequately support the main claims. Since these experiments cannot be completed within the revision timeline, we believe withdrawal is the most responsible option.

**Withdrawal Confirmation:**

I have read and agree with the venue's withdrawal policy on behalf of myself and my co-authors.